

# FG-Droid: Grouping based feature size reduction for Android malware detection

Recep Sinan Arslan

Department of Computer Engineering, Kayseri University, Kayseri, Turkey

## ABSTRACT

**Background**. The number of applications prepared for use on mobile devices has increased rapidly with the widespread use of the Android OS. This has resulted in the undesired installation of Android application packages (APKs) that violate user privacy or are malicious. The increasing similarity between Android malware and benign applications makes it difficult to distinguish them from each other and causes a situation of concern for users.

**Methods**. In this study, FG-Droid, a machine-learning based classifier, using the method of grouping the features obtained by static analysis, was proposed. It was created because of experiments with machine learning (ML), deep neural network (DNN), recurrent neural network (RNN), long short-term memory (LSTM), and gated recurrent unit (GRU)-based models using Drebin, Genome, and Arslan datasets.

**Results**. The experimental results revealed that FG-Droid achieved a 97.7% area under the receiver operating characteristic (ROC) curve (AUC) score with a vector including only 11 static features and the ExtraTree algorithm. While reaching a high classification rate, only 0.063 seconds were needed for analysis per application. This means that the proposed feature selection method is faster than all traditional feature selection methods, and FG-Droid is one of the tools to date with the shortest analysis time per application. As a result, an efficient classifier with few features, low analysis time, and high classification success was developed using a unique feature grouping method.

## INTRODUCTION

Android OS is a mobile platform that was prepared by a group of developers. It has dominated the mobile operating system market for many years. According to 2021 statistics, it is used in more than 70% of mobile devices, and together with the iOS operating system, it meets 98% of the entire market share (https://www.statista.com/statistics/272698/global-market-share-held-by-mobile-operating-systems-since-2009/). Accordingly, the number of application downloads worldwide is increasing rapidly and this trend is expected to continue (https://www.statista.com/statistics/266488/forecast-of-mobile-app-downloads/). There are many reasons for this widespread use of the Android operating system. It has the appropriate functional infrastructure to access hardware resources. It is free and open source platform and is equipped with a security framework that relies on the Linux kernel (*Khanna & Singh, 2016*). However, since its security structure is based

Corresponding author
Recep Sinan Arslan,
sinanarslanemail@gmail.com

on the application layer (*Smalley & Craig, 2013*), these devices become partially or completely exposed to numerous security attacks, making them a routine target (*Wang & Li, 2021*). In addition, the widespread use of mobile devices with Android OS, the adoption of these devices by end users and the resulting of increasing market share, has caused it to become the target of cyber hackers, especially web-based and application layer-based attacks (https://www.businessofapps.com/data/android-statistics/; https://securelist.com/it-threat-evolution-q1-2021-mobile-statistics/102547/).

When malicious applications access user mobile devices, they can engage in a series of malicious activities, such as obtaining confidential information, seizing more authorized user accounts, and misusing the obtained certain level of security (*Wang & Li, 2021*). Google Play has set up a permission-based system to control applications' access to confidential data. Users are asked to give permission before installation, taking into account the resources of the application. Users must approve these permissions before installation to use it. However, this-preapproval mechanism does not provide sufficient protection for users, since users accept these permissions without detailed examination. As a result, users accept all conditions in order to have free access to applications that offer the features they demand (*Ratibah Tuan Mat et al., 2021*). For these reasons, there is a need to work on the security mechanism that Google Play provides for users.

Different types of malware detection mechanisms have been proposed to address this need in the security mechanism. These are the signature-based approach (*Sihang et al., 2020*; *Rahman et al., 2018*), behavior-based detection software (*Su et al., 2020*; *Saracino et al., 2018*; *Chen et al., 2021*), and machine learning-based approaches (*Nguyen Vu & Jung, 2021*; *Lachtar, Ibdah & Bacha, 2021*; *Sihang et al., 2021*). Among these, machine learning and deep learning architectures have gained more popularity recently. Because in this method, it is possible to obtain both a dynamic learning and development process and good results against zero-day attacks. It is possible to create ML models that are open to learning and development at the same speed for cyber attackers who try to overcome the security mechanism with a new technique every day, and to produce promising solutions for the detection of malware (*Ou & Xu, 2022*). In machine learning based approaches, there is a dependency on the features used as input, classifier and learning architectures. In this study, the size of the feature set and the effects of the features it contains in terms of performance and efficiently in android malware detection mechanisms are focused.

There are many manual and automatic feature extraction methods. These methods are basically divided into three groups, as static features, dynamic features, and hybrid features (*Handrick da Costa et al., 2020*). Each of these features can be used to detect different malicious activities. In addition, each feature can provide a distinctiveness in detecting malicious applications. For this reason, different feature sets can be used in studies in this field, and problem-specific feature vectors can be produced. The main purpose is to provide fastest and highest classification success model with the best feature set.

The contributions of this work in the following way:

- Feature grouping based Android malware detection tool (FG-Droid) is a low runtime and highly efficient machine learning model.
- The model groups permission-based features with a unique methodology and obtains only 11 static features for each application.
- The model has 97.7% classification success in the tests and only needs 0.063 s for analysis per application. This value is one of the best values among the models with similar classification success. This value was obtained without using the GPU.
- The model selects fewer features than traditional feature selection methods (chi2, f_class_if, PCA) and requires less processing time while showing higher classification success than them.
- As a result, a model with high classification success and low analysis time with few features has been revealed thanks to the proposed unique grouping.

The continuation of this study is organized as follows: In 'Materials & Methods', Android application development infrastructure and similar studies on this subject are analyzed separately for static and dynamic methos. The methodology of FG-Droid is explained in detail in 'Results' and the experimental results are given comparatively in 'Conclusions'. In the last part, a general evaluation of the study is made and suggestions are made for future studies. In addition, a FG-Droid permission grouping table is given as Appendix A before the reference section.

# MATERIALS & METHODS

## Literature review

In this section, recent studies related with Android malware detection, feature generation and selection, and static, dynamic, and hybrid approaches are discussed.

## Static analysis

The features extracted as a result of the static analysis are basically obtained from the Androidmanifest.xml in the APK package SMALI files. It is intended to extract features without actually running applications and use them for classification. Permissions, application programming interface (API) calls, intent filters, application metadata, function calls, and opcode are some of these features. Among these, permission and API calls are more commonly used.

Permissions are one of the most researched and widely used features in malware analysis because they form the basis of the android security architecture. Android applications are installed on mobile devices after approval from the users. Since permission is the first obstacle for cyber hackers to reach their malicious targets, many researchers (*Sinan Arslan, Alper Doğru & Barışçı, 2019*; *Shehata et al., 2020*; *Thiyagarajan, Akash & Murugan, 2020*) have carried out permission-based analysis studies. *Arp et al. (2014)* evaluated the permissions, API calls, hardware components, and intents in the application manifest file together in their study in 2014. They classified the obtained static feature vector with support vector machine (SVM). APK Auditor (*Kabakus, Doğru & Çetin, 2015*) is static analysis tool running on a central server with signature database based on application

analysis. When the system was tested with 6909 samples, it was able to detect malware with 88% accuracy. The specificity value was 0.925. *Syrris & Geneiatakis (2021)* examined malware detection based on machine learning and static analysis. The six best-known ML techniques were evaluated in terms of both the classification rate and feature selection. In the experimental results, it was shown that a high accuracy rate could be achieved with a lower dimensional feature set, as in FG-Droid. Anastasia (*Fereidooni et al., 2016*) is a classifier using machine learning (decision tree (DT), random forest (RF), SVM, *etc.*) and a deep neural network. The static analysis tool was created to extract features such as intent, system commands, permissions, and API calls. In the tests performed with a dataset consisting of 11,187 benign and 18,677 malicious applications, a TP value of 97.3% was obtained and the f-score was 96.0%. Droidmat (*Wu et al., 2012*) is another tool that uses permissions, intents, API calls, and services to classify android APKs. Analysis was performed using the K-neighbors algorithm. As a result, 91.82% f1 score was obtained. Drebin (*Razgallah et al., 2021*) works with a large dimensional static feature vector, since limited hardware resources pose problems for dynamic analysis. The extracted feature vector was trained with SVM and a classification success of 94% was achieved. It was stated that an average of 10 s was required for the analysis.

API calls is another feature that can be obtained as a results of static analysis. Because applications need API callas to communicate with the device. Therefore, API calls can be useful to understand the intent of applications. Many researchers (*Jung et al., 2018*; *Pektaş & Acarman, 2020*; *Sharma & Dash, 2014*; *Alazab et al., 2020*) have targeted malware detection using API calls. High classification success has been achieved in studies with API calls, but there are approximately 32,000 different APIs on the Android platform (*Ou & Xu, 2022*). This causes the feature vector to be very large size. In addition, a very limited part of these features were used by applications. DroidAPIMiner (*Aafer, Du & Yin, 2013*) uses package-level parameters in the feature vector to catch malicious API calls. As a result, a 2.2% FP rate was achieved. MAMADroid (*Onwuzurike et al., 2019*), which analyzes the API usage behavior of applications, is a more robust system for malware detection. It was reported that the model has shown a high success rate in tests for many years. Taheri et al. (*Taheri et al., 2020*) similarly used the feature vector produced by using API calls, intents, and permission from training and prediction phases for 4 different classifiers. Similarity calculation was made using Hamming distance and an accuracy value of 91% was achieved. The static analysis method has been used in many other studies, such as DroidSieve (*Suares Tangil et al., 2017*) and DroidDet (*Zhu et al., 2018*).

Although, static analysis is advantageous at certain points, is also has some limitations (*Bakour & Murat Ünver, 2018*). It would not be possible to observe its behavior at runtime. Code analysis of complex software takes time. It may not be possible to extract static features based on the source code inn encrypted and obfuscated applications.

A summary of all studies using the static analysis methodology used in this study is shown in Table 1.

**Table 1  Summary of current works on static analysis based in Android malware detection.**

| Ref | Dataset | Feature extraction | Classification | Classification rate | Sec. for identification each app |
|---|---|---|---|---|---|
| Arp et al. (2014) | Drebin | Used permissions, sys. Api calls, network address | Machine learning | 94% | 10 |
| Anastasia (Fereidooni et al., 2016) | Own dataset | Api calls, network address | ML(NB, RF, KNN) | 96% | 0.29 |
| MamaDroid (Onwuzurike et al., 2019) | Drebin | Api calls, call graphs | SVM, RF, 1-NN, 3-NN | 87% | 0.7 ± 1.5 |
| Taheri et al. (2020) | Drebin, Genome | Api calls, intents, permissions(21492 features) | FNN, ANN, WANN, KMNN | 90%–99% | Very high |
| Apkauditor (Kabakus, Doğru & Çetin, 2015) | Own dataset | Permissions, services, receivers | Signature based | 92.5% | – |
| Syrris & Geneiatakis (2021b) | Drebin | Static features | ML(6 six classifiers) | 99% | – |
| Droidmat (Wu et al., 2012) | Own dataset | Intents, Api calls | Signature based | 91.83% | – |
| Alazab et al. (2020) | Own dataset | Api calls | ML (RF, J48, KNN, NB) | 94.30% | 0.2 –0.92 |
| Pektaş & Acarman (2020) | Drebin, AMD, Androzoo | Api calls | SDNE (DNN model) | 98.5% | – |
| Shehata et al. (2020) | Own dataset | Activities, services, receivers, providers, permissions | RF | 97.1% | – |
| Thiyagarajan, Akash & Murugan (2020) | Androzoo | Permissions(113) –>PCA (10) | DT with PCA feature selection | 94.3% | – |
| **Proposed Model** | **Drebin, Genome, Arslan** | **Permission groups** | **ML, DNN** | **97.7%** | **0.063** |

## Dynamic analysis

Dynamic features are extracted as a result of analyzing the situations in which Android applications communicate with the operating systems or the network. System calls and network usage statistics are the most basic features that give an idea about the APKs. In addition, processor and memory space usage data, the status of the services running instantaneously, some statistical data about the device (battery usage, screen-on time, *etc.*), and information about the addresses reached by the systems calls and network packets are important features for classification. Droidscope (*Kwong Yan & Yin, 2012*) is an efficient and effective dynamic Android malware detection tool that works on Android devices by extracting three-layer (hardware, operating system, and dalvik virtual machine) system calls, and performing semantic analysis at the operating system and code level. MADAM (*Dini et al., 2012*) tracks kernel-level system calls and user-level usage statistics and activities to be

able to describe and classify the behavior of a mobile application. ANDLANTIS (*Bierma et al., 2014*) is a dynamic analysis tool that runs on a sandbox. It aims to detect malware by analyzing system calls, footprints, and running behaviors. It requires 1 h for the analysis of 3000 applications. (*Chen et al., 2017*) proposed a semi-supervised classifier that works using dynamic API usage logs. They made use of both labeled and unlabeled data to obtain application properties. They showed the results comparatively by classifying with SVM and k-nearest neighbor (KNN). TaintDroid (*Enck et al., 2014*) is a dynamic tracer tool. It uses Dalvik virtual machine to do this tracking. It tracks the usage of sensitive resources such as the location, microphone, and camera. While revealing the traces of the applications due to the virtual machine, it does not pose any danger on the real environment. This ensures that data leaks are prevented and malicious software intentions are revealed.

Dynamic analysis-based approaches try to understand the intentions of mobile applications by analyzing their behavior during running. For this reason, in some cases, it can show higher recognition success than static analysis. However, in order to understand a suspicious behavior, it must be run at least once and information must be collected during this time. This means both creating a security problem for the device and additional processing time (*Bala et al., 2021*). The presence of processor and memory limitations of mobile devices complicates the applicability of dynamic analysis. However, running applications on an emulator/sandbox may be sufficient for dynamic analysis. It was accepted that it is possible to gather more information about the application by simulating the so-called user behavior (*Alzaylaee, Yerima & Sezer, 2020*). Crowdroid (*Iker, Urko & Nadjm-Tehrani, 2011*) is a behavior-based android malware detection tool that works in client–server architecture. All system calls from the application are collected via the mobile device and sent to a cloud server for analysis. With K-means, this data is processed and the application is classified.

A single feature can capture certain aspects of an application. However, using more than one feature together can be more advantageous in malware detection. Various studies have been conducted with hybrid feature structures using combinations of both static and dynamic features (*Surendran, Thomas & Emmanuel, 2020*; *Martin, Lara-Cabrera & Camacho, 2019*; *Tong & Yan, 2017*). ProfileDroid (*Wei et al., 2012*) evaluates both static and dynamic features such as Android permissions, features obtained as a result of code analysis, and network usage statistics. Thus, a systematic study was aimed to establish a cost-effective and consistent model.

## Hybrid analysis

The features obtained because of static and dynamic analysis alone can capture only one aspect of the applications. For this reason, it is possible to make a more accurate analysis when more than one feature group is used together. Thus, it is possible to detect malware with higher accuracy. NTPDroid (*Arora & Peddoju, 2018*) uses a hybrid feature vector that combines permissions and network traffic. The FP-Growth algorithm is proposed to obtain the commonly used thicknesses among applications. Experimental results showed that it has an accuracy value of 94.25%. *Arshad et al. (2018)* proposed a model for Android malware detection using static and dynamic analysis together. The model first extracts the

requested permissions, used permissions, application components and suspicious API calls by examining the APK file. Then, the application's network usage statistics obtain system calls as features. By combining these two feature groups, the feature vector of the application is obtained and classified by SVM. Experimental results showed high malware detection accuracy. OmniDroid (*Martín, Lara-Cabrera & Camacho, 2019*) is a hybrid feature vector and machine learning classification tool that combines static and dynamic features based on voting. In this study, the dependencies of static and dynamic features are taken into account and the features go through an evaluation mechanism when combining them into a single vector. AAsandbox (*Bläsing et al., 2010*) offers a two-stage analysis approach. First, an image of the application is taken in offline mode and static and dynamic analysis is performed on the sandbox. The entire system is hosted on the cloud server and suspicious applications are detected. *Tong & Yan (2017)* proposed a different approach as a solution to the long processing time problem of static analysis and the problem of dynamic analysis to consume a lot of processing resources. The model can detect different types of malware much more efficiently than other studies and achieves an average of 90% classification success.

In the hybrid analysis process, there are models that use static and dynamic features together while creating the feature vector, and there are studies in which hybrid classifiers are used to classify a single feature type. *Yerima, Sezer & Muttik (2014)* proposed a composite classification model to increase classification accuracy. In the composition classifier, rule-based, function-based, tree-based, and probability-based classifiers are used together. Thus, an average of 5% increase in success was achieved.

## Methodology

In this section, the method used by the FG-Droid tool was explained in detail. The method includes the stages of extracting features, grouping extracted features, updating feature values, and testing on sample dataset.

## Android application structure

Applications prepared for the Android OS are presented to users as a kind of compressed file with the APK extension. The package includes files such as source code, manifest.XML, libraries, resources, DEX file, and properties, as shown in Fig. 1. Applications are developed in Java using the Android SDK. Applications whose source code is completed are converted to Dalvik bytecode (DEX) together with other required files. A manifest file is an XML that contains basic cookies for the applications, links to external files, and libraries such as activities, receivers, and content providers. In addition, the permissions needed to access device resources, target platform data are included in the manifest. External resources such as videos, sounds, audios, images, and text files used by the application are packaged in the APK file. As a result, a package is prepared that contains necessary files to execute the entire working functions (*Syrris & Geneiatakis, 2021*). Prepared APK packages are offered to users via Google Play Store or third party application distribution platforms. More details for Android application development are provided on the developer page (https://developer.android.com/docs).

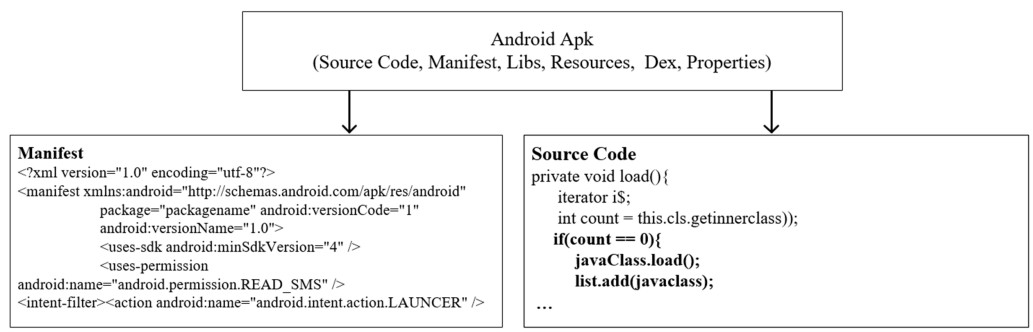

**Figure 1** Structure of Android Apk (*Syrris & Geneiatakis, 2021a*).

## Automatic feature extraction and pre-processing

The Android operating system works on an application-based. After the applications are prepared, they are made into zip files, and their extensions are determined as APK. These applications, which have similar file structures, can be run on the same operating system (Android OS). The application package contains folders and files such as androidmanifest.xml and class.dex, resources.arsc, lib, res. Androidmanifest.xml is required for applications to run and contains some information (version, API level, hardware, software information, *etc.*) and permissions declared by the application.

This study adopts a static analysis-based feature extraction approach for Android malware detection. A series of steps were applied to extract the permission-based features in the Androidmanifest.xml file, as shown in Fig. 2. First, a dataset consisting of benign and malicious software was created. Applications are opened using the Jadx decompiler. Thus, access to both the source code and the Androidmanifest.xml file needed in this study was provided. A list of all permissions in the Androidmanifest.xml file is required to determine the permissions requested by the applications and transfer them to the feature vector. This study used a list consisting of 348 permissions in Android 11 API level 30.

The permissions in the manifest file and the permissions in the permission list were compared and transferred to the feature vector with a value of 1 for used ones and 0 for unused ones. Thus, a feature vector of $1 \times 349$ dimensions was obtained for each application. It should be noted that it is impossible to open all applications healthily with reverse engineering. In this case, values such as NaN and space in the vector may be present. These were checked, and related applications were filtered. There was no problem in terms of malicious application labelling applications. However, it is critical to determine whether benign apps are genuinely benign. For this reason, all benign applications were rechecked on Virustotal. For this reason, high-speed and efficient feature extraction and preprocessing were carried out. Those who did well in these tests were marked with B, and applications from the Drebin and genome dataset were marked M. After these processes were completed for all applications, they were combined into a single feature vector. As a result, a feature vector of $7,266 \times 349$ dimensions was obtained. The resulting vector was saved in a CSV file and converted into a usable form in the following steps. All these processes were done with the exe file prepared in c# language.

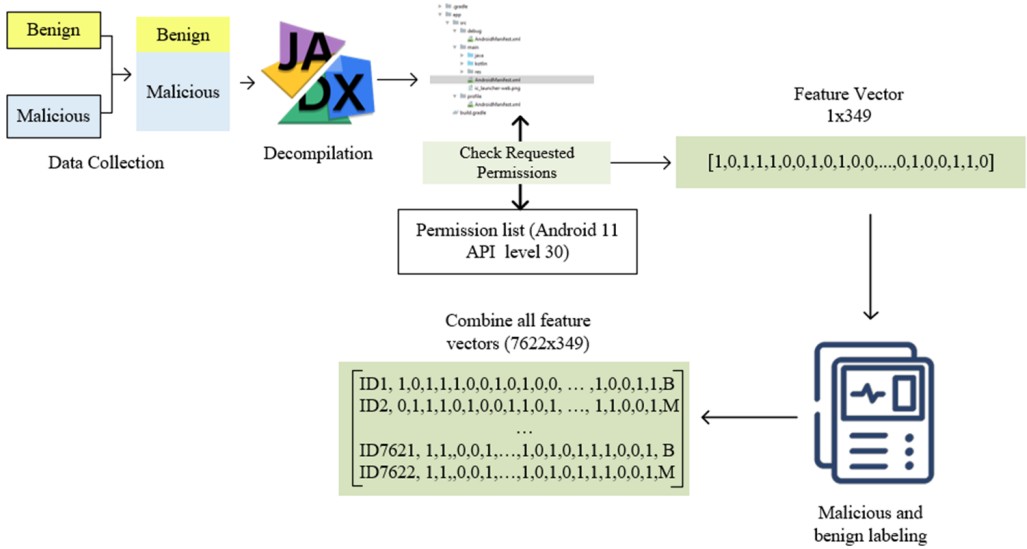

**Figure 2** Automatic feature extraction and pre-processing flowchart.

Considering that there will be a need for high processing power in this large-sized vector in machine learning models, it is subjected to feature grouping, the details of which we have given in the next section. Since the values after the group are greater than 1, normalization is performed with the normal distribution, and all values are scaled to the 0–1 range.

## Proposed feature grouping and feature selection algorithm

The mechanism of accessing certain components or performing functions was based on permissions in the Android operating system. Android applications request permission to access and read information about calendar, location information, contacts, storage, camera, microphone, various sensors. For example, it is possible to access contact information on the device with the Android.permission.READ_CONTACT permission. Permissions vary according to the device's feature and functional capacity.

In this study, instead of a feature vector containing all permissions used in Android architecture, a model called FG-Droid has been developed, which can achieve high classification success with a very fast training and testing time by using fewer features. For this purpose, a series of operations were carried out within the flow chart shown in Fig. 3.

In the first stage of the model, a dataset containing 7,622 applications was created, the details of which are given in Section 4. The permission-based features of these applications have been extracted. At this stage, a training and test vector containing 349 features was created. In order to shorten the training and testing times and to classify using less processing time, operations were carried out according to the proposed algorithm for grouping the features and reduction the size. Thus, a tool was created in which it was possible to classify with less features. The standardization process was performed using the normal distribution on the feature vector obtained after this step. The resulting data

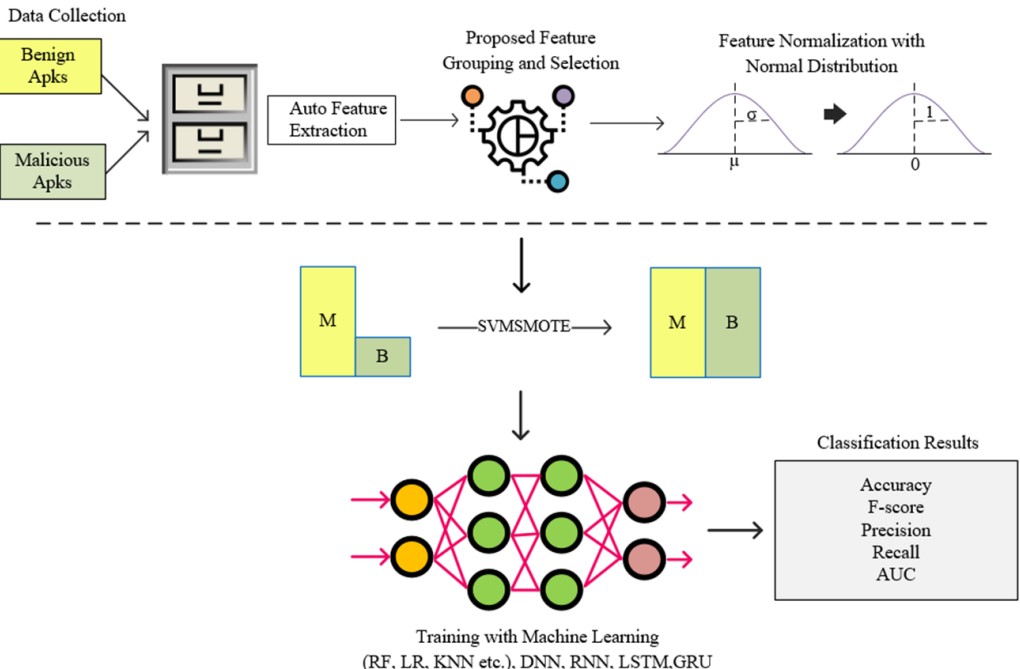

Data Collection

**Figure 3** **Flow-chart of proposed model.**

set was divided into two separate parts to be used in the training and testing phases. During the training phase, oversampling was performed using the SVM-synthetic minority over-sampling technique (SMOTE) algorithm to resolve the sample number imbalance between clusters, and training was carried out on deep neural network (DNN), recurrent neural network (RNN), long short-term memory (LSTM), and gated recurrent unit (GRU) networks together with machine learning algorithms (Rf, Extratree, gradient boosting, *etc.*). The resulting trained models were tested with previously separated test data and the results were shown in comparison with commonly used metrics.

As can be seen in Fig. 3, the proposed feature grouping and selection method was carried out after the application features are extracted. Thus, it was possible to work with a much lower-dimensional feature vector not only in the training and testing phases, but also in all processing steps. The details of feature grouping operations are as shown in Fig. 4.

The determination of groups was based on basic read/write operations (CRUD) in computer systems and operations specific to mobile devices (Broadcast, Control, Bind). These groups consisted of Access (A), Modify (M), Set (S), Update (U), Write (W), Read (R), Get (G), Manage (Mn), Bind (Bd), Broadcast (B) and Control (C). The Android permissions in each group are shown in Appendix A. Within the groups determined in Appendix A, all of the features were first scanned and brought together on a group basis. The values of these features were aggregated within each group, so that the existence of the permission represents itself in the total. For example, instead of using 21 different features in model trainings separately, they were combined under the Access (A) feature. Thus, 21 properties were represented only by the Access (A) property. Although there are

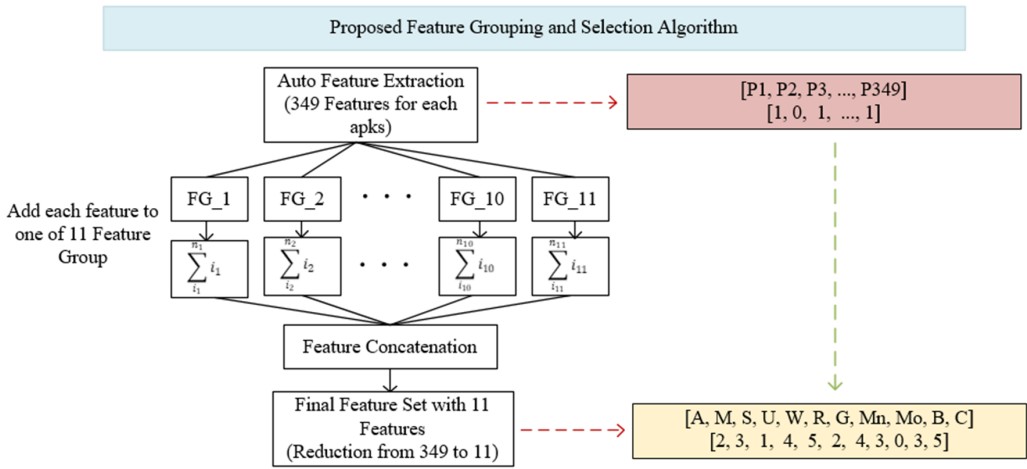

**Figure 4  Proposed feature grouping and selection algorithm.**

---

Algorithm: **Feature Grouping**
**Input:**    originalfv : 2-dimensional array, fgroup: 2-dimensional array
**Output:** groupedfv: 2-dimensional array

row = len(originalfv)
col = len(originalfv[0])

**for** i ← 0 to row
    **for** j ← 0 to col
        **for** k ← 0 to 11 {
            **if** originalfv[i][j] **in** fgroup[k] **then**
                groupedfv[i][k] += originalfv[i][j];
        }
**return** groupedfv;

---

**Figure 5  Pseudo-code of feature grouping and selection algorithm.**

---

many permissions/features in the Android operating system, very few of them were used in applications, so it is not necessary to use all of the features separately during training and testing. In the Drebin dataset used in this study, the average number of permissions requested for each application was five. Similarly, the average number of permit requests in the Genome and Arslan datasets was six and five, respectively. In this case, a feature vector created with five numbers 1 and 344 numbers 0. Instead, all of the permissions were searched for each application, and they were combined under 11 feature groups after grouping and aggregation. Thus, in the process starting as 349 features, a feature vector with only 11 features was obtained for each application.

The feature grouping process was carried out using the code structure given in Fig. 5. The feature vector of 7,622 × 349 dimensions taken as input was converted into a 7,622 × 11 dimensional vector, thus providing a much more efficient learning in training and testing processes. The effect of the obtained vector on the results is shown in detail in 'Results'.

## RESULTS

The performance of the proposed model is evaluated in terms of (1) the classification success and (2) the time needed for training and prediction. A binary classification was determined as benign and malicious. The tests were performed separately on machine learning techniques (RF, DT, LDA, *etc.*), DNN, RNN, LSTM, and GRU networks, and the results were given in detail. Thus, it was desired to show the effect of feature grouping and reduction process used by FG-Droid application on the results.

### Evaluation metrics and experimental setup

The experimental environment and evaluation criteria are very important to analyze the performance of the machine-learning model. For the experiment, 70% of the data was split for training and 30% for testing. area under the receiver operating characteristic (ROC) curve (AUC), precision, recall, f-score, and accuracy values were calculated to evaluate the performance of the proposed model. The calculation equations of these metrics are shown below:

$$\text{precision} = \frac{TP}{TP + FP} \tag{1}$$

$$\text{recall} = \frac{TP}{TP + FN} \tag{2}$$

$$\text{f-score} = \frac{\frac{TP}{TP+FN} * \frac{TP}{TP+FP}}{\frac{TP}{TP+FN} + \frac{TP}{TP+FP}} \tag{3}$$

$$\text{accuracy} = \frac{TP + TN}{TP + TN + FP + FN} \tag{4}$$

TP is the number of truly malicious samples from those predicted as malware, FP is the number of samples that are predicted to be malware that are not truly malicious. FN is the number of samples that are predicted to be benign that are not truly benign and TN is the number of truly benign samples from those predicted as benign.

The computer system architecture used in the development of the FG-Droid tool is as shown in Table 2. It is very important in calculations regarding training and test times. All comparative results were obtained using the same infrastructure.

### Dataset

In studies on Android malware detection, it is not possible to reach sufficiently large, homogeneously distributed, and reliable datasets. In this study, Drebin (https://www.sec.tu-bs.de/~danarp/drebin/) and Genome (http://www.malgenomeproject.org/) datasets were used for malware applications. The Drebin dataset contains 5,560 samples from 179 different application groups. A malicious dataset with 6,660 samples was created by taking

**Table 2  Computer system architecture.**

| Parameter | Value |
| --- | --- |
| CPU | Intel Core i5-10760 |
| RAM | 8 Gigabyte |
| Operating System | Windows 10 Pro |
| Python version | 3.7.6 |
| Libraries | Scikit learn, matplotlib, pandas, seaborn, numpy, imblearn |

1,000 samples from the genome dataset. The Arslan (*Arslan, Alper Doğru & Barışçı, 2019*) dataset was used for benign applications. In this dataset, the applications with the highest number of downloads in Google Playstore were selected in various categories, and they were subjected to security tests on virustotal.com (https://www.virustotal.com/gui/home/upload) and a data set consisting of 960 applications that passed the test was created. During the labeling phase of the datasets, Drebin and Genome were distributed as labeled, so no action was taken on the malicious dataset. However, labeling in the benign dataset was made by us to be used in this study. As a result, a dataset containing real-world applications and examples that will not create noise for both malicious and benign applications has been created.

## Hyper-parameter tuning for best machine learning models

In order to achieve successful classification performance in a test environment where each sample is represented by 11 features, 10 different machine learning techniques and DNN, RNN, LSTM, and GRU networks were designed and tested. At the end of these processes, hyper-parameter tuning was performed for the all classifiers.. Both GridSearch and RandomSearch algorithms were used for the selection of the best parameters, and the selection range of the parameters and selected parameters was as shown in Table 3. Thus, it was possible to obtain the best rates for all classifiers in the tests.

## Results for machine learning

After the feature-grouping algorithm used in the development process of the FG-Droid, the results obtained in the tests using machine-learning techniques were as shown in Table 4 for 10 different classifiers.

As can be seen in Table 4, the accuracy rate is 90% and above, except for two algorithms, and the highest classification rate was 92.5%. While a successful result such as 94.4% was obtained in the precision value, recall, and f-score were 92%. As a result, a very high success of 97.9% was achieved in the AUC score, which is the indicator of classification success for both classes. This showed that it is not enough for permission-based applications to remove the disruptive features in android malware detection and it is not necessary to consider each permission as a separate feature. It was possible to reach the 97.9% success level using only 11 features of the application. The effect of the FG-Droid tool is not in the feature selection, but in grouping the features and ensuring that each permission is represented under the group. It was possible to prevent the loss of the distinctiveness of the

**Table 3  Details of the hyper-parameters of all classifiers.**

| Classifier | Hyper-parameter tuning range | Selected values |
|---|---|---|
| KNeighbors | n_neighbors: (1,20,1)<br>p: (1,5,1)<br>weights: ('uniform', 'distance'), | 'n_neighbors': 5<br>'p': 3<br>'weights': 'uniform' |
| SVC | C: [0.1, 1, 10, 100, 1000]<br>kernel: ['rbf','linear'] | C: 1,<br>kernel: 'rbf' |
| GradientBoosting | loss: ['log_loss', 'deviance', 'exponential']<br>"learning_rate": [0.01, 0.025, 0.2] | loss:log_loss<br>"learning_rate": 1.0 |
| Random Forest | n_estimators: [100, 200, 800, 2000]<br>criterion: ["gini", "entropy", "log_loss"] | n_estimators=100<br>criterion='gini' |
| XG Boost | 'max_depth': [3, 5]<br>'learning_rate': [0.01, 0.1, 1, 10], | max_depth=5<br>'learning_rate'=1.0 |
| Extra Tree | 'n_estimators': [200-2000]<br>max_features': ['auto','sqrt','log2']<br>max_depth: [10-110]<br>Min samples split: [2,5,10] | 'max_features': sqrt<br>'n_estimators': 1800<br>max_depth:30<br>Min samples split:10 |
| Ada Boost | 'learning_rate': [0.01, 0.1, 1, 10],<br>'n_estimators': [50, 500, 2000] | 'learning_rate': 1<br>'n_estimators': 50 |
| Decision Tree | 'criterion': ['gini', 'entropy'], | criterion='gini' |
| Logistic Regression | 'C': [1.e-03, 1.e-02, 1.e-01, 1.e+00, 1.e+01, 1.e+02, 1.e+03],<br>'penalty': ['l1', 'l2']} | C': 1.0,<br>'penalty': 'l2' |
| Linear Discriminant Analysis | solver: ['svd', 'lsqr', 'eigen'] | 'solver': 'svd' |

**Table 4  ML classification results with proposed feature grouping algorithm.**

| ML classifier algorithm | AUC_score | Precision | Recall | F-score | Accuracy |
|---|---|---|---|---|---|
| KNeighbors | 0.965 | 0.905 | 0.913 | 0.909 | 0.910 |
| SVC | 0.950 | 0.844 | 0.931 | 0.885 | 0.881 |
| GradientBoosting | 0.965 | 0.928 | 0.899 | 0.913 | 0.916 |
| Random Forest | 0.976 | 0.942 | 0.904 | 0.923 | 0.925 |
| XG Boost | 0.976 | 0.941 | 0.905 | 0.923 | 0.925 |
| Extra Tree | 0.979 | 0.944 | 0.909 | 0.926 | 0.929 |
| Ada Boost | 0.955 | 0.917 | 0.872 | 0.894 | 0.898 |
| Decision Tree | 0.970 | 0.939 | 0.901 | 0.920 | 0.922 |
| Logistic Regression | 0.881 | 0.790 | 0.816 | 0.803 | 0.803 |
| Linear Discriminant | 0.856 | 0.769 | 0.814 | 0.791 | 0.788 |

feature by selecting the feature. Thus, both the number of features were greatly reduced and the effect of many features on classification was used under the group.

In order to improve the results, the cross-validation process on the model was performed using "RepeatedStratifiedKFold" function. The n_splits, n_repeats and random state were chosen as 10, 3, and 123, respectively. The results obtained in repeated tests for all classifiers are shown in Fig. 6. Accordingly, XGB, ET, RF and DT algorithms are more successful than other classifiers at the average classification rate.

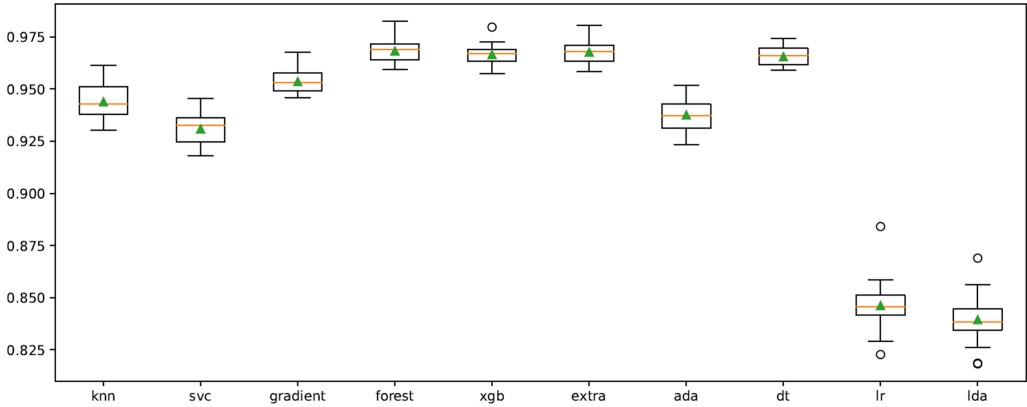

**Figure 6** Cross-validation graph for all ML classifiers.

## Results for deep learning models

During the development of the FG-Droid, tests were carried out using deep learning models. As stated before, the number of features decreases considerably with the proposed grouping algorithm. The effect of the algorithm in deep learning models, which need to use more features and more training examples in its basic structure, is very important. These tests were carried out to observe whether the group-based feature vector would have a negative effect on the classification performance in deep learning models.

As can be seen in Table 5, a lower classification success was achieved when compared to the machine learning models, due to the reduction in the number of features. This decrease occurred for all of the metrics. The highest level of success was achieved with the LSTM (100,100) model with 92.2% for accuracy. The precision, recall, and f-score values were 94.4%, 91.6% and 93.9%, respectively. The highest value of 92.5% was obtained in the AUC score, in which both classes were evaluated together. The learning curves for the models with the highest classification rate for DNN, RNN, GRU, and LSTM are shown in Fig. 7. For all of the models, learning took place very quickly, with training reaching its peak at 20 epochs. The test curve was parallel to the training curve. However, the learning rate slowed down after 20 epochs. This slowdown was thought to be due to the need for more data to continue learning. Repeating the tests with a larger dataset will allow FG-Droid to achieve a higher AUC.

## Comparison results and best classifier results details

In the FG-Droid development process, it was understood that the models in which the proposed algorithm produced the most successful results were based on the machine learning algorithms. Machine learning techniques have generally shown high success.

In order to evaluate the results with all classifiers, ROC curves were taken as shown in Fig. 8. Accordingly, 97.0% and above AUC values were obtained in random forest, ET, DT, and XGBoost algorithms. Obtaining a high classification value in different classifiers strengthens the widespread effect of the proposed feature grouping approach. The highest

**Table 5** Deep learning models classification results with proposed feature grouping algorithm.

| Deep learning models | Auc_score | Precision | Recall | F-score | Accuracy | Total number of processed parameters | Epoch |
|---|---|---|---|---|---|---|---|
| DNN(30,30) | 0.918 | 0.936 | 0.906 | 0.921 | 0.925 | 1352 | 50 |
| DNN(30,30,30) | 0.920 | 0.927 | 0.916 | 0.921 | 0.921 | 2282 | 50 |
| DNN(30,30,30,30) | 0.935 | 0.942 | 0.908 | 0.925 | 0.930 | 3212 | 50 |
| DNN(100,100,100) | 0.925 | 0.944 | 0.891 | 0.917 | 0.924 | 31702 | 50 |
| DNN(300,300,300) | 0.928 | 0.940 | 0.904 | 0.921 | 0.926 | 275102 | 50 |
| RNN(10,10) | 0.900 | 0.910 | 0.910 | 0.920 | 0.902 | 242 | 50 |
| RNN(30,30) | 0.916 | 0.928 | 0.891 | 0.939 | 0.918 | 1322 | 50 |
| RNN(100,100) | 0.867 | 0.899 | 0.830 | 0.863 | 0.875 | 11402 | 50 |
| RNN(300,300) | 0.887 | 0.905 | 0.865 | 0.885 | 0.895 | 94202 | 50 |
| GRU(10,10) | 0.910 | 0.918 | 0.897 | 0.907 | 0.908 | 712 | 50 |
| GRU(30,30,30) | 0.902 | 0.915 | 0.887 | 0.901 | 0.905 | 3932 | 50 |
| GRU(100,100) | 0.915 | 0.836 | 0.891 | 0.913 | 0.913 | 34102 | 50 |
| GRU(300,300) | 0.914 | 0.934 | 0.893 | 0.913 | 0.915 | 282302 | 50 |
| LSTM(10,10) | 0.920 | 0.939 | 0.891 | 0.914 | 0.918 | 902 | 50 |
| LSTM(30,30) | 0.909 | 0.890 | 0.908 | 0.908 | 0.910 | 5102 | 50 |
| LSTM(100,100) | 0.916 | 0.940 | 0.890 | 0.914 | 0.920 | 45002 | 50 |
| LSTM(300,300) | 0.916 | 0.937 | 0.892 | 0.914 | 0.920 | 375002 | 50 |

value of 97.7% was obtained with the ET classifier. This value represents high malware detection. Achieving high classification success with only 11 features is valuable.

## The effect of proposed feature grouping on learning and testing time

The proposed feature grouping-based algorithm performed a very large feature vector size reduction and feature selection from 349 features to 11 features. Thus, the amount of data was reduced by 30 times and a much simpler feature vector was obtained. This had a serious impact on the training and test duration as well as on the classification result. Having hardware limitations in mobile devices and the need for fast and efficient tools are other advantages in choosing FG-Droid.

In the initial state of the created feature vector, in the case of using well-known feature selection methods and in the tests made with the proposed feature grouping method, the required times for training and testing were as shown in Table 6.

The results obtained when using known and feature selection methods, such as Extratree, randomforest, chi2, f_classif, f_regression, and PCA, or without feature selection, are as shown in Table 6. The FG-Droid tool reached a 97.7 AUC with only 11 features. The minimum number of features required to obtain similar AUC values was 35. It was seen that the increase in the number of selected features had a serious effect on both the training and prediction time. FG-Droid was on average 700% faster in the training time than in the model without any feature selection, while it was approximately 80% faster in the prediction time. On the other hand, the best results were obtained with chi2 among the traditional feature selection methods, and the proposed model was 45% faster in the training time and 23% faster in the testing time when compared to the chi2 method. As a result, by grouping

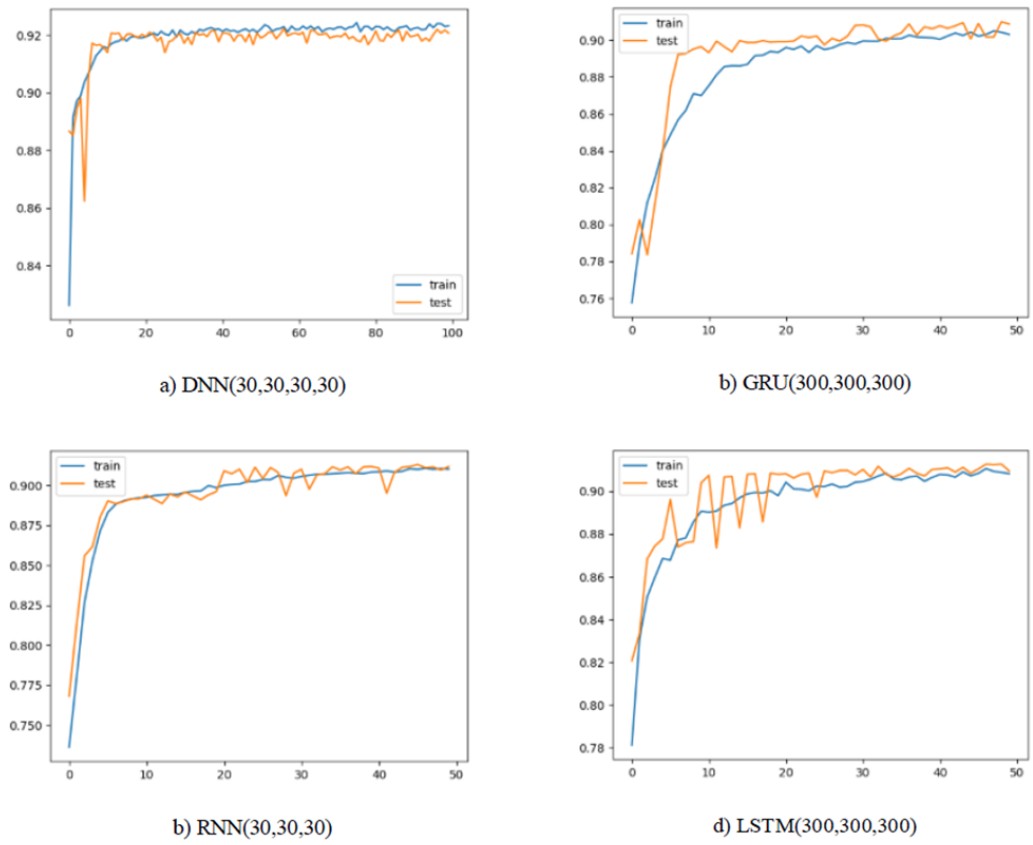

Figure 7 (A–D) Learning curves of deep learning models.

the features with FG-Droid, the effect of the feature on the classification was not lost and an efficient classification was made.

## Comparison with similar works and discussion

A lot of work has been done in recent years on Android malware detection. These studies basically used static analysis, dynamic analysis, or hybrid feature extraction methods. While some of these obtained features contributed positively to the classification performance, some may have had no effect at all, and some may have had a deteriorating effect. For this reason, it is beneficial to determine those features that contribute positively to the result and to remove the others from the feature set. In this study, a feature grouping method was proposed for the use of features extracted from static analysis in the classification. FG-Droid used the feature vector consisting of grouped features and made classification with the ExtraTree algorithm. Instead of selecting and removing features from the feature vector, the approach of evaluating these features within the group was adopted. The Drebin malicious dataset, which has been widely used for many years, was used in the tests for FG-Droid.

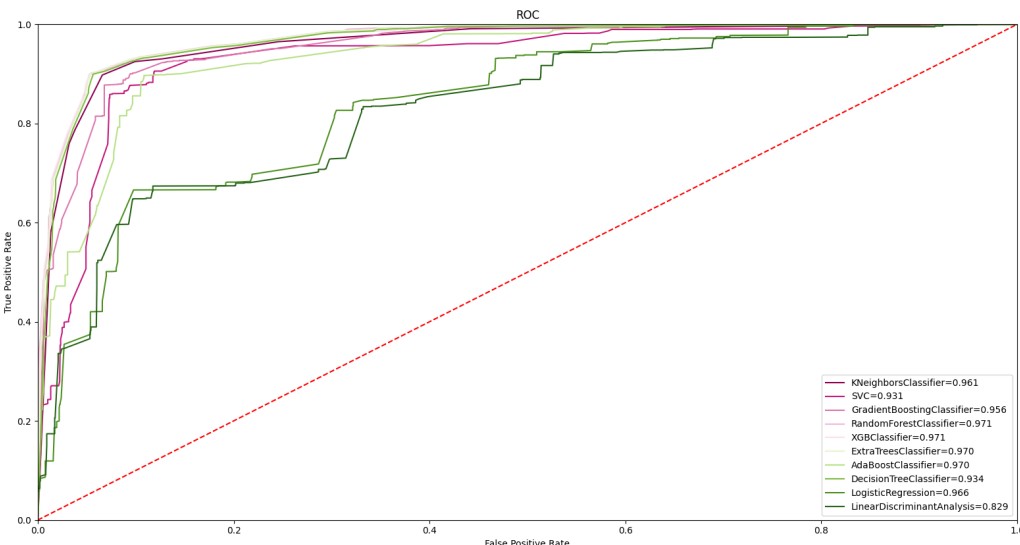

**Figure 8** ROC curve of all ML algorithms with feature grouping.

**Table 6** Training and testing time with different feature selection algorithms and proposed model.

| Feature selection method | Selected/Grouped feature size | Training time(s.) | Prediction time(s.) |
| --- | --- | --- | --- |
| FG-Droid (proposed model) | 11 | 0.344 | 0.063 |
| Without feature selection | 349 | 2.234 | 0.109 |
| Feature importance with extra tree algorithm | 46 | 0.609 | 0.078 |
| Feature importance with random forest algorithm | 45 | 0.594 | 0.078 |
| chi2 | 35 | 0.500 | 0.078 |
| f_classiif | 35 | 0.519 | 0.078 |
| f_regression | 35 | 0.516 | 0.078 |
| PCA | 35 | 0.688 | 0.078 |

The results of recent studies and the test results of the FG-Droid are shown in Table 7. When the studies were examined, it was understood that Drebin was widely used as a dataset, and in some studies, Androzoo and original datasets were studied. Permissions have generally been the most widely used feature in static analysis-based methodologies. These features were used in the training and testing processes of various classifiers. Classification successes vary between 91.0% and 99.0%. In the analysis time per application, the best value was 0.008. When the studies were compared, it was understood that the tool with the highest classification success and the best analysis time per application was the FG-Droid recommended in this study. In the study conducted in 2021 with an analysis time of 0.008, the classification performance remained at the level of 91.0%. The analysis time of FG-Droid of 0.063 s reveals that it is a very efficient model. The most important factor in the emergence of this efficient model is of course working with a low number of features. It works with a lower feature count than all the studies given in the table. In almost all

**Table 7 Similar works with proposed model.**

| Paper | Dataset | Feature extraction | Feature selection or grouping algorithms and classification methods | Classification Performance | Sec. for identification each app |
|---|---|---|---|---|---|
| *Ratibah Tuan Mat et al. (2021)* | Androzoo, Drebin | Permissions | Naïve Bayesian | 91.10% | – |
| *Mohammed Arif et al. (2021)* | Androzoo, Drebin | Permissions | MCDM | 90.54% (4 classification levels) | – |
| *Millar et al. (2021)* | Drebin, Genome | Opcode and permissions | No feature selection, classification with multi-view deep learning | 91.00% | 0.008 |
| *Zhang et al. (2021)* | Drebin | Text sequences of apps | No feature selection. Classification with Text CNN | 95.20% | 0.28 |
| *Arp et al. (2014)* | Drebin | Used permissions, sys. Api calls, network address | Machine learning | 94% | 10 |
| Anastasia (*Fereidooni et al., 2016*) | Own dataset | Api calls, network address | ML(NB, RF, KNN) | 96% | 0.29 |
| MamaDroid (*Onwuzurike et al., 2019*) | Drebin | Api calls, call graphs | SVM, RF, 1-NN, 3-NN | 87% | 0.7 ± 1.5 |
| *Taheri et al. (2020)* | Drebin, Genome | Api calls, intents, permissions(21492 features) | FNN, ANN, WANN, KMNN | 90%–99% | Very high |
| Apkauditor (*Kabakus, Doğru & Çetin, 2015*) | Own dataset | Permissions, services, receivers | Signature based | 92.5% | – |
| *Syrris & Geneiatakis (2021b)* | Drebin | Static features | ML(6 six classifiers) | 99% | – |
| Droidmat (*Wu et al., 2012*) | Own dataset | Intents, Api calls | Signature based | 91.83% | – |
| *Alazab et al. (2020)* | Own dataset | Api calls | ML(RF, J48, KNN, NB) | 94.30% | 0.2 –0.92 |
| *Pektaş & Acarman (2020)* | Drebin, AMD, Androzoo | Api calls | SDNE(DNN model) | 98.5% | – |
| *Shehata et al. (2020)* | Own dataset | Activities, services, receivers, providers, permissions | RF | 97.1% | – |
| *Thiyagarajan, Akash & Murugan (2020)* | Androzoo | Permissions(113) ->PCA (10) | DT with PCA feature selection | 94.3% | – |
| **FG-Droid** | **Drebin, Genome, Arslan** | **Permission groups** | **ML, DNN** | **97.7%** | **0.063** |

studies, feature selection is avoided and the processing time is compromised in order to achieve high classification success. However, limited resources in mobile devices require consideration in efficiency. In this study, the feature selection approach was handled from a different perspective and instead of selecting features, a joint evaluation approach was adopted.

## CONCLUSIONS

The increase in mobile devices using the Android operating system has caused them to be the target of cyber attackers. New types of malware are emerging every day, and new methods have been proposed as a precaution. FG-Droid uses a permission grouping-based approach to Android malware analysis. It has an AUC of 97.7% with 11 features for binary classification. Using this newly proposed algorithm, 349 features extracted from Android applications were grouped and reduced to 11 features. Thus, a much more efficient feature vector was revealed. Drebin and Genome malware datasets were used to observe the effect of the model. With the success of the classification, a very efficient model was created with the shortening of the training and prediction times. In the future, tests will be performed with datasets containing more samples to further increase the classification success. Analysis time per application is just 0.063 s, one of the best analysis times ever. In addition, since a very fast method has been developed, it is aimed to present it with a platform that will serve online. As a result, FG-Droid is expected to contribute positively to the security of Android smart devices.

### Funding
The authors received no funding for this work.

### Competing Interests
The authors declare there are no competing interests.

### Author Contributions
- Recep Sinan Arslan conceived and designed the experiments, performed the experiments, analyzed the data, performed the computation work, prepared figures and/or tables, authored or reviewed drafts of the article, and approved the final draft.

### Data Availability
   The raw data, features for grouped permissions, are available in the Supplementary File.

### Supplemental Information
Supplemental information for this article can be found online at http://dx.doi.org/10.7717/peerj-cs.1043#supplemental-information.

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
