# Peer review of "FG-Droid: Grouping based feature size reduction for Android malware detection"

_PeerJ Computer Science, doi:10.7717/peerj-cs.1043_

## Round 0.1 · original submission · Major Revisions

The detailed comments are as follows, please carefully address all the comments before re-submission.

Reviewer 1 ·

Basic reporting

The English language used in the paper is not up to publishable academic standards. The abstract is written in the past tense. Apart from numerous grammatical mistakes, there are many spelling mistakes in the paper. What makes it worst, the text written from lines 174 to 177 is in some other language (possibly Turkish).
The above-mentioned mistakes prove that the author did not bother to proofread the paper before submission. This is unacceptable.
The literature review skeleton is good. However, a lot of language mistakes make it difficult to follow. Moreover, a comparison table of the reviewed papers should be included.
Figure 5 (Pseudo Code): It appears that the Pseudocode is generated through some automated tool. Convert it to write it in standard algorithm notations. The current form is unacceptable,

Experimental design

Needs to be reconsidered

Validity of the findings

The author proposes a reduced feature set (using just 11 static features) that is enough for classification tasks for Android malware and Benign apps. Moreover, the author claims that there is no need for an extended feature set for the classification task. I have some concerns in this regard:
1) Most of the literature published in the Android malware detection domain use significantly big feature sets for malware detection and classification tasks. The proposed techniques (I can not mention a single here because there are loads of them) have claimed up to 99% detection accuracy by using larger feature sets. Moreover, a lot of them are using fairly big datasets as compared to this study. How would you compare your results with them?
2) Another concern is about adversarial evasion attacks. A small feature set is more vulnerable to evasion attacks than compared to a diverse feature set. Have you considered it?
3) I can see that this study uses around 6k malware and 900+ benign samples. How can you justify this distribution? Normally, balanced datasets are employed. However, in real-world situations, malware samples are always less than benign, therefore, an unbalanced dataset (more benign and less malware still makes sense (As used in Drebin)).
4) You have used Drebin and Genome datasets. Just to let you know that all the malware samples in Genome are already a part of the Drebin dataset. Therefore no need to use the Genome as a separate dataset.
5) Drebin dataset is now obsolete. The apps in Drebin were gathered from 2010-to 2012. That's a decade old now. Please consider new datasets. A good option would be to use KronoDroid (2021).
6) Figure 8 is an example of poor presentation of results.

Additional comments

Although the paper should be rejected, however, I am suggesting a major revision at this stage point. All my concerns should be addressed one by one in rebuttal for further considereation.

·

Basic reporting

The English language should be improved to ensure that an international audience can clearly understand your text. Their are too many grammatical mistakes, the article need to go through proofreading
Introduction should be rewrite again
Missing sentences in Line 16
This has resulted in the undesired installation of Android apks that violate user privacy or malicious ???
Line 32 it -> It, Also, too many use of 'it' in paragraph
Line 45 ->When malicious applications Access user mobil devices - correct it
Line 51 ->suffiecient -> correct it sufficient
Line 55 -> Malware detection mechanism in different types have been proposed to address this (WHAT IS THIS?? IN THIS SENTENCE)need in the security mechanism. These (?) are signature-based approach ...
Line 79 wit -> with

Experimental design

Line 238 While adding to the csv file, malicious and benign applications were labeled. Please explain How the data is labeled in Figure 2
Line 241 The processing steps shown in Figure-2 were carried out by means of an automatic code, and it
is a very fast feature extraction process. How this process is fast? please explain in detail
Automatic Feature Extraction and Pre-processing section need to be explain properly. Things are overlapping and explained shortly.

Validity of the findings

Explain how FG-droid is efficient as mentioned in contribution?
Line 427 -> While some of these obtained features contribute positively to the classification performance, some may have no effect at all, and some may have a deteriorating effect. Explain How? What are their limitations? What have you achieved?

Additional comments

In this study, FG-Droid, a machine-learning based classifier, using the method of grouping the
features obtained by static analysis, was proposed. It was created because of experiments with Machine
learning (ML), DNN, RNN, LSTM and GRU based models using Drebin, Genome and Arslan datasets.


However, Clear, unambiguous, professional English language is missing throughout the paper. Overall the structure of the paper need to be modified again. Experimental evaluation with other studies are missing in terms of novelty.

---

## Round 0.2 · accepted · Accept

Congratulations, the reviewers are satisfied with the revised version of the manuscript and have recommended the acceptance decision.

Reviewer 1 ·

Basic reporting

I strongly recommend the authors to re-write the last sentence in the abstract. The word "thanks to" should be replaced. Apart from that, the paper satisfies the publication standards.

Experimental design

Good

Validity of the findings

Justified

Additional comments

The author has put a lot of effort to enhance the quality of the work. I am happy to accept the paper for publication.